# New Policy Research on Education Development and Global Citizenship in a Sustainable Environment

Hanlin Fang [1], Fengrong Zhang [1,*], Qianwen Xiao [2] and Ciyun Lin [3]

1. College of Marxism, Northeast Normal University, Changchun 130024, China
2. Department of Economic Management, Ningbo Institute of Engineering, Ningbo 315211, China
3. School of Transportation, Jilin University, Changchun 130012, China
* Correspondence: zhangfr631@nenu.edu.cn

**Abstract:** At present, multinational and regional educational agencies and researchers have used academic development and global citizenship training to increase globalization. However, owing to the existence of emerging policies, these innovations have raised issues in the educational sector. Therefore, in this paper, EP-SE has been proposed for enhancing the sustainable environment in education and global citizenship. EP-SE seeks to elucidate the educational developments and reveal the part played by non-discursive components in ensuring global citizenship cooperation. The emerging policies for sustainable development emphasize the importance of moral ideals, personal accountability, and engaged global citizenship. Capitalist debate impartially privileges a real economy based on the individual and enhanced income. EP-SE has been developed within the current commodity conditions, critical democracy, and threats to educational development and can serve as a model for sustainable global citizenship development. The experimental results suggest that the proposed model achieves the highest performance compared with the other existing methods.

**Keywords:** sustainable environment; critical democracy; global citizenship; EP-SE





## 1. Introduction to Sustainable Educational Development and Global Citizenship

The primary objectives of education are to train people within society, enhance their credentials for economic employment, and incorporate people into the community [1]. Education is a way to make people socialize and allow society to run smoothly and safely. Education is one of the main drivers of growth [2]. Education increases efficiency and innovation and encourages entrepreneurship and technical progress [3]. Moreover, it plays a significant role in ensuring economic and social growth and improving income distribution. By mixing with those from different cultures and backgrounds, children develop a healthy social contact [4]. Increases in positive energy, personality, and trust can contribute to the child's development. Those that receive an education are able to earn higher wages, have more opportunities and appear to be happier in their lives [5]. Societies with high education levels have less criminality, better health in general, and better political participation [6]. The main cause of poverty is regarded as a lack of access to education that facilitates the acquisition of expertise, skills, and experience, both in professional life and in personal life. In addition, education can improve trust by developing additional communication skills and help people to solve problems and achieve their goals [7]. Education provides us with awareness of knowledge, specific skills, and ethics that can be used to improve the world. Undoubtedly, good hard work is necessary for success in life; however, it does not produce results without education [8].

This program aims to educate students on the principles, behaviors, and attitudes that encourage responsible global citizenship: creativity, innovation, and dedication to peace, human rights, and sustainable development [9]. A global citizen is someone who understands the broader world and recognizes their position within it and can play an active

cultural role in making the world more peaceful and sustainable. The idea of global civics training has begun to replace or overlap with trends within the education system, including multicultural education, peace education, human rights education, sustainable development education, and international education [10,11]. Even if the idea of global citizenship is not recent, in a modern, growing, and increasingly interconnected world, it is becoming ever more critical [12]. Critical thinking requires the capacity to challenge one's view of the world and exercise empathy and appreciation for other cultures [13]. It includes respect for others, compliance with rules and regulations, and setting an excellent example [14]. Global citizens feel a sense of duty to help when the rights of others are violated, regardless of where in the world the situation occurs [15]. Local action will generate global change when people join others [16]. The concept of a global citizen rests on two fundamental assumptions: first, that global culture is changing or expanding, and people can then use these changes to define themselves: second, that there is a bright future for international people. Global citizenship has advantages resulting from increased knowledge and the challenges associated with global problems: alien encounters, through which people are more aware of the difficulties that various parts of the world face; broadened horizons; and the building of skills and understanding [17]. Citizenship training provides children with awareness and the ability to understand challenges relating to politics, the media, civil society, the economy and the law, and to engage with democracy [18]. Sustainability encourages an environment where waste and pollution are reduced, emissions are reduced, employment is increased, and resource distribution is improved [19]. Education for sustainable development does not mean that it is eco-friendly; that requires changes in lifestyle, including in leadership, communication, and management [17].

The main contributions of this paper are:

(1)     By addressing the challenges of a sustainable environment and educational development and the concept of global citizenship, this paper presents an efficient and successful approach to calculating the EP-SE system of education and global citizenship for a sustainable environment. The differentiation of education is illustrated, providing a non-dialectical component focused on the functioning of global citizenship and emphasizing the concept of personal responsibility or ethics associated with global citizenship. It enhances the sustainability of teaching and learning and shows that globalization, which depends on commodity conditions, can be enhanced.

(2)     Different, distinct and invariant principles are considered in education for sustainable environment and global citizenship. Textual, statistical, performance, predictive, and discriminant analyses were conducted by working with the C-ESD, AGCE, E-GC, TGCE, and AME-SD methods proposed in the literature review. The experimental paper collected about 4245 tables divided into four levels of six features to perform the calculations. The implementability of the EP-SE policy in this paper was finally verified.

The structure of this paper is as follows: Section 1 introduces the ideas of sustainable educational development and global citizenship. Section 2 discusses various aspects of existing research on sustainable academic development and global citizenship. Section 3 develops the proposed emerging policies for a sustainable environment. The evidence that validates the findings is discussed in Section 4. Finally, Section 5 addresses the outlook for the future.

## 2. Literature Review

Even if the concept of global citizenship is not new, it is becoming increasingly important in a modern, developing, and increasingly interconnected world.

Wahyudin introduced the C-ESD, of which the following have been examined in this research: (1) policies and practices to implement the new curriculum, providing more space for peace education within ESD; (2) indigenous aspects of the support for peace and development in the context of sustainable education; and (3) schools focusing on peacebuilding in the context of sustainable development education [20].

Alida proposed an AGCE and discussed the crucial role of integration in the promotion of the AGCE. Significant global policy progress towards integration and how developments in inclusive childhood education can benefit AGCE were summarized. The paper ends with an innovative two-fold practice for childhood training, worldwide competence, and critical literacy, which offers possibilities for contributing to AGCE in the form of the conceptual and physical aspects of access [21].

MacQueen et al. introduced E-GC; this analyzes the degree to which the continuing problem of promoting essential citizenship is tackled, locally and internationally, in the curricula. They evaluated selected Australian primary school curriculum documents, with Australia as the limitation, to assess the degree of engagement in global citizenship education. Although well-intentioned, more work is necessary to ensure their findings are applicable to other schools [22].

Karen et al. proposed TGCE for a globalized economy; most TGCEs have been identified within the liberal world, and there is confusion over the various TGCE 'forms'. New interfaces between neo-liberal–liberal, liberal–critical, critical, and critical–post-critical neo-conservative have been described. Although TGCE is very diverse, the paper claimed that the typologies of TGCE remain largely framed within a restricted variety of possibilities, particularly in the modern-colonial imagination [23].

Mark et al. proposed AME-SD for each field of focus; in conjunction with the intent of this study, it examines the specifics of instructional approaches, tracking, and assessment methods, and outcomes have been recorded in the evaluated publications. Their analysis can therefore provide helpful advice for AME education or pedagogical practices and for the development of monitoring and assessment instruments for AME-SD [24].

Based on the literature survey of the existing C-ESD, AGCE, E-CG, TGCE, and AME-SD educational development and non-discursive components described above, sustainable development education and global citizenship are common challenges. To overcome these challenges, the newly proposed technique, EP-SE, was used. This demonstrates the differences in education, provides non-discursive components focused on global citizenry operations, and underlines the personal responsibilities or moral concepts associated with global citizenship. Pedagogical sustainability, with regard to global dependence on commodity conditions, can be increased.

## 3. Emerging Policies for Sustainable Environment

Environmental sustainability is characterized as responsible engagement with the environment to ensure that natural resources are not depleted or degraded and to ensure long-lasting ecological quality. Environmental sustainability is responsible for preserving natural resources and protecting global habitats, both now and in the future, to enhance sound health [25–27]. With regard to ecological sustainability, sustainable development—helping the environment, eliminating use, maintaining and preserving habitat and the entire environment—is often the main focus. Researchers believe that sustainable practices should be integrated into a service's operations. The sustainable development idea is linked to environmental protection. It brings a new way of designing and developing, which considers economic, social, and environmental issues. Sustainable growth is achievable growth that an enterprise can sustain without problems. The highest growth rate a business can hold without raising its financial leverage is the sustained growth rate [28,29].

It has been indicated that certain methods are more suitable in the context of the fall of the welfare state and new views on globalization. The EP-SE will allow us to look beyond old barriers to separate civic (as a form of social science) education and global education (which emphasizes political activity beyond the study of political science or the practice of community involvement). Different national theories (liberal, republican and cosmopolitan) form the basis of directions of movement for justice, equality, and sustainable development. Evidence from international student curricula vitae suggests that students often react to a specific type of global reality that is considered by educational policymakers

to exist. Researchers propose consideration of what globalization is, how it is understood, and the unstated political imperative behind it. The concept of globalization is generally questionable as it results from several different political, economic, and cultural forces. For example, the two dominant subjects are about responding economically and culturally to a particular context, without considering alternatives [30–32].

As is evident in Figure 1, sustainable development education promotes the awareness, skills, understanding, values, and behavior necessary to build a sustainable global environment that protects and preserves the environment, promotes social justice, and fosters economic sustainability. Quality education focuses on the entire child, including gender, race, ethnicity, socioeconomic status and geographic location, social, emotional, behavioral, physical, and cognitive development. It makes the child ready for life rather than for objective testing. Instead, it seeks to ensure fair access to affordable training, remove gender and wealth disparity and ensure that quality higher education is universally accessible. Transforming pedagogy requires committed learning. It uses Pablo Freire's theories, such as the dialogical approach, and therefore, "bank schooling". It is often an effective way to prevent knowledge that lacks depth and, hence, gain a more profound understanding. Transcultural education is how specialized healthcare deals with the idea of community. It is a mental discipline in care delivery that concentrates on global culture and comparative education, health, and treatment. This field of inquiry and practice was formally established in 1955. Conflict reduction is a way of resolving a dispute between two or more parties under negotiation. Arguments may be individual, financial, political, or emotional. When a conflict occurs, arbitration to settle the dispute is always the best course of action [33].

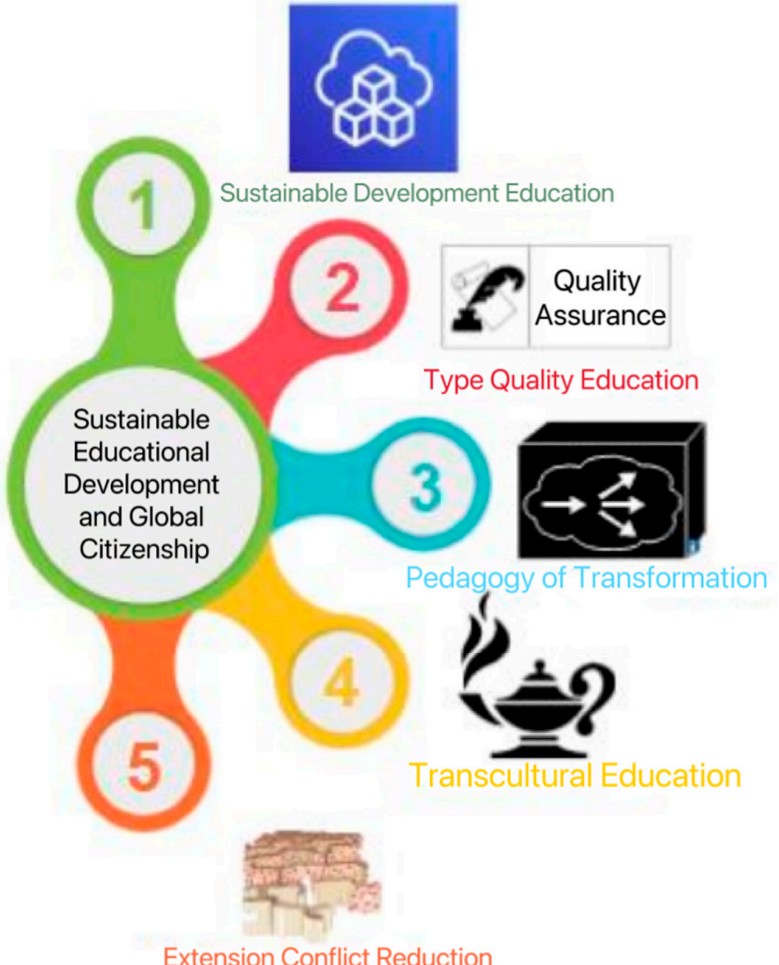

**Figure 1.** Values of sustainable education and global citizenship.

As shown in Figure 2, the curriculum is the backbone of the learning process, in that the curriculum supports the creation of the courses offered, instructional materials, student learning plans and evaluations, and even teacher education. In addition to sharing objectives among teachers and students, the curriculum standardizes the learning objectives for the whole school, providing students with consistent paths from one grade to the next. Simply put, sustainable growth is realizable growth that can be maintained without problems by a company from the context of a company. The highest growth rate that can support a business without increasing financial leverage is the sustainable growth rate. Global social issues (worldwide problems) have never been limited to national borders and need to be addressed nationally and internationally. A strong example of this mechanism is atmospheric pollution. Internalization takes place when an individual carries out a transaction themself, instead of involving someone else. This concept can be applied to enterprises, investments, or environmental enterprises. Internalization in business is not an open market transaction; therefore, it represents an internal market within a company [34].

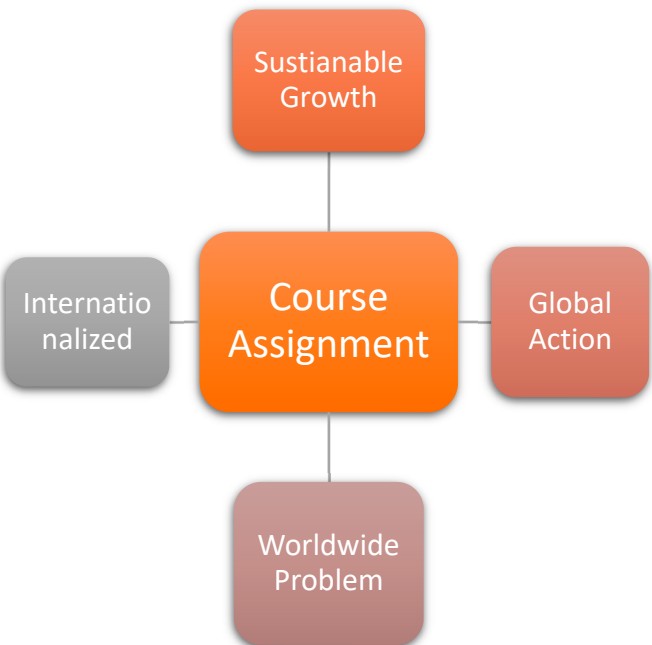

**Figure 2.** Contribution for education development and global citizenship.

The development of environmental sustainability and learning is shown in Figure 3. With regard to global citizenship, environmental information is primarily incorporated for data transmission and retrieval. Every group collaborated on environmental information and multi-control services regulates all strategies for sustainable development. Some improvements were made to the newly introduced EP-SE framework in the developing global education system. Finally, environmental and sustainability planning was undertaken.

Global citizens feel responsible for helping when the rights of others are violated, regardless of their location in the world. Local action can create global change when people join together. Environmental information should contain all the information about the zero conditions of environmental elements and the interactions between such elements. The interrelated elements have been compiled under five factors: transport and accessibility, conservation of natural assets, environment quality, 'social assistance', and 'high density of use'. This research also includes the factor of implementation.

$$A_{xy} = \beta \frac{GCD_x^{\vartheta_1} \times GCD_y^{\vartheta_2}}{B_{xy}^{\vartheta_3}} \tag{1}$$

Equation (1) describes the skill volume of global citizenship: the traditional model of velocity, which relates skill volumes and distance between countries to their economic scales. The conventional skill model between member organizations, $x$, $y$, is as shown in Equation (1). The global citizenship education of countries, $GCD_x$, $GCD_y$, is calculated, where $A_{xy}$ is the skill volume between $x$ and $y$ areas. $B_{xy}$ is the gap between $x$ and strong $y$ areas; $\beta$, $\vartheta_1$, $\vartheta_2$, and $\vartheta_3$ are the parameters to be evaluated.

The different types of multifunctional input, along with the weighted total of this information, is shown in Figure 4. The weights of all numerical inputs are multiple, and the input is weighted. The performance is ultimately calculated as in Equation (2).

$$C = g(h_1 e_1 + h_2 e_2 + f) \tag{2}$$

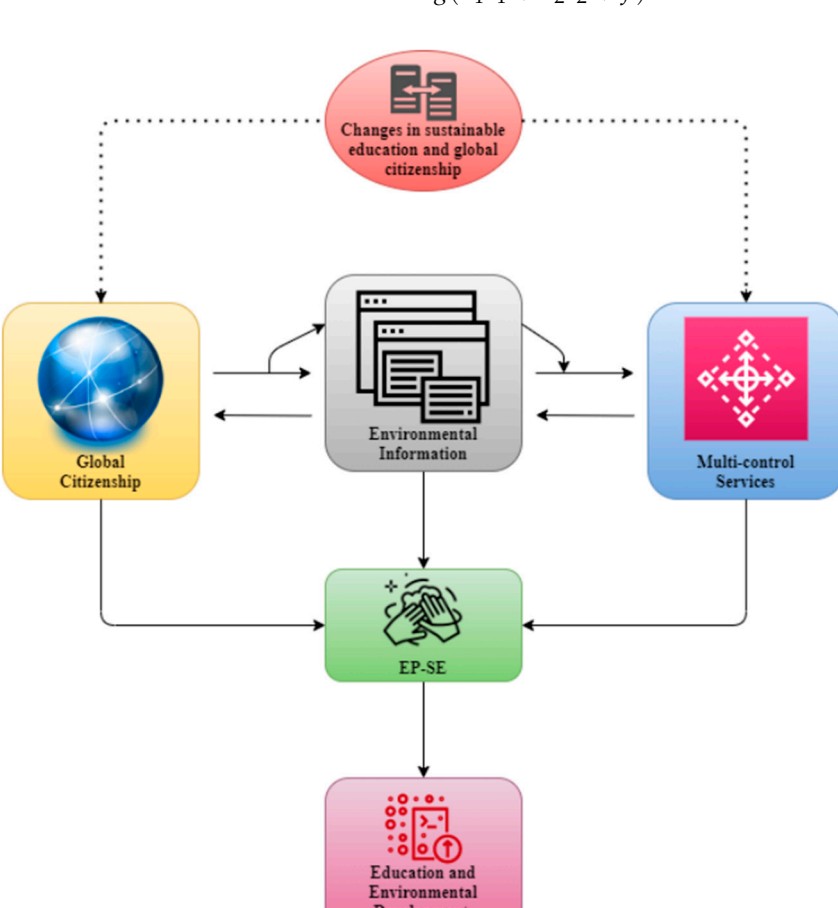

**Figure 3.** Development of environmental and educational sustainability.

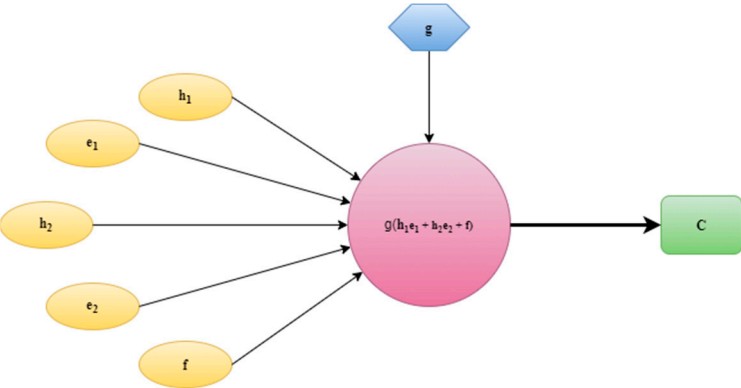

**Figure 4.** Non-linear depiction.

From Equation (2), the performance analysis techniques have increased. Each input has a corresponding weight, $h$, assigned to other inputs on account of its relative significance. The node uses a function, $g$, as a weighted sum of its information, shown in Equation (2). $e_1, e_2$ , $f$ (called bias) are numbering inputs and $h_1$ and $h_2$ are weights. As indicated in Equation (2), the output $C$ is calculated. The $g$ function is never linear and is known as activation. Such activation functions are aimed at introducing non-linearity into neuron production. It is essential because most data from the real world are nonlinear, and we want neurons to learn these nonlinear depictions.

$$e_t = \frac{e_1 - mean\ (e)}{std\ (e)} \tag{3}$$

The transaction flows of development education are denoted in Equation (3). Before starting the analysis report, many information collection strategies have been used. First, all entries which do not have any transaction flow value, $e_t$, are removed since these are economically irresistible, $e_1 - mean\ (e)$. Second, linear reverse and forward interpolation are used to extrapolate missing data, $std\ (e)$.

$$\sigma = \frac{1}{M} \sum_{x=1}^{M} e_x \tag{4a}$$

$$\mu = \sqrt{\frac{1}{M} \sum_{x=1}^{M} (e_x - \sigma)} \tag{4b}$$

The calculation of $\sigma$, the distributed standard, when $\mu$ is the sampling distribution, and is the number of observations defined as both, is shown in Equation (4a,b).

$$N^2 = 1 - \frac{TT_{res}}{TT_{tot}} = 1 - \frac{\sum_x (f_x - f)^2}{\sum_x (f_x - f)^2} \tag{5}$$

The coefficient of determination, $N^2$, illustrates the model's expectations as shown in Equation (5), and variance in the vector table to offer flow based on independent characteristics can be expected from the individual characteristic space variables. Here, $TT_{res}$ is the residual square count, $TT_{tot}$ is the total square amount or ratio to data variance and $f_x$ the expected input vector of $f_x - f$ and the average data observed. In Equation (5), the text analysis techniques have increased.

$$N^2 = 1 - \frac{TT_{res}}{TT_{tot} + \omega} \tag{6}$$

In Equation (6), the statistical analysis is increased when $N^2$ is an empirical parameter which has been set and it varies between 0 and 1: When $N^2 = 0$, the model still does not predict the target variable, and the model will predict the target variable perfectly if $N^2 = 1$. Any value from 0 to 1 indicates the percentage of the target variable features that can be explained using the model. When $N^2 < 0$ , the model is no better than one that continuously forecasts the mean of the destination variable.

The error function shows whether specific estimates differ from the target. The weights are adjusted according to this amount during the learning phase. The error function decreases, as seen in Figure 5, when the network is training.

$$MSE = \frac{1}{M} \sum_{x=1}^{M} (f_x - f)^2 \tag{7}$$

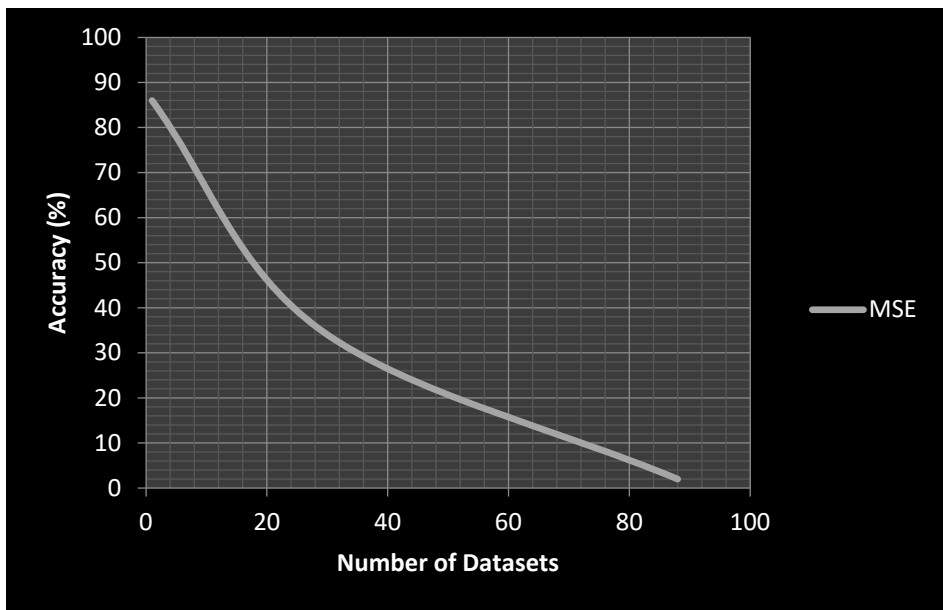

**Figure 5.** Graphical representation of MSE.

*MSE* indicates how far the predictions deviate from the target via the mean square error expressed in the above Equation (7). The weights are revised in this quantity during the training process. The loss function decreases during network training. If the actual observation is $f_x$, $f$ is the expected observations, and $M$ is the number of observations. Root medium RMSE error is often seen as it is mainly used to measure the contrast between values predicted using a model or estimator with observed values.

$$\frac{dH}{dS} = p\left(1 - \frac{H}{W}\right)H \tag{8}$$

From Equation (8), the predictive analysis techniques are increased. Let us assume that the population will expand as a derivative based on the logistic model. The calculation model can be written as in this case, where $p$ is consistent, $H$ is the population at time $S$, and $W$ is the environmental carrying power.

$$\frac{dH}{dS} = p\left(1 - \frac{H}{W}\right)H - tH = g(H) \tag{9}$$

The $g(H)$ classification and regression changes to the above Equation (9) when harvests occur and a proportion of the population is eliminated. When $p$ is a constant population growth rate, $t$ is the proportion of the population being harvested, $H$ is the population at $S$, and $W$ is the environmental supporting capability.

$$\frac{dH}{dS} = p\left(1 - \frac{H}{W}\right)H - L \cong g(H; P; W; L) \tag{10}$$

where $r$ is constant economic growth, $L$ is the number of people harvested from the population, $H$ is the population at time $S$, and $W$ is the environment's ability to support; the $H$ population can be calculated at constant harvest as in Equation (10).

$$L = p\left(1 - \frac{H}{W}\right)H \tag{11}$$

It is important to sequence information to ensure a stable balance for citizens. In this case, understanding how the economy can reach balance is complicated, and the calculation

of a parameter, such as $H$, is complex. Resolution of $L$ in Equation (11) followed by the optimization of the right-hand side of the equation can be used to calculate the optimal sustainable yield.

$$\frac{d(H - W)}{dS} \cong -P(H - W) \tag{12a}$$

$$H(0) = H_0 => H - W = (H_0 - W)q^{-ps} \tag{12b}$$

Discriminant analysis techniques are increased due to the above Equation (12a,b). The magnitude of the retrieval time of $H$ to its transport capability, $W$, follows the disturbance of $W$ to a certain extent because $P - W$ is small. The time for recovery after a minor irritation from $W$ is substantial ($H(s) - W$).

$$S_N(L = 0) = \frac{1}{P} ln\left(\frac{H_0 - W}{H - W}\right) \tag{13}$$

It has been found that the solution depends heavily on the value of $H_0 = W$. Similarly, the time of recovery, $S_N$, for its carry-force is the same after a minor disturbance of $W$, with different values, and $L = 0$.

An efficient and successful EP-SE is therefore set out in this paper. It illustrates the variation in education, provides non-discursive components focused on global citizenry operations, and highlights the personal responsibilities or moral concepts associated with global citizenship. It enhances pedagogical sustainability and shows that globalization dependent on commodity conditions can be increased.

## 4. Results and Discussion

A method for calculating the sustainable environment of an EP-SE system in education and global citizenship is presented in this paper and the data were provided by a prominent data provider. Table 1 shows details of the dataset.

**Table 1.** Details of the data used in the experiment.

| S. No | Data | Description |
|---|---|---|
| 1 | User count | 1, 5, 10, 15, 20, ..., 50 |
| 2 | Sessions for users | 10, 20, 30, 40, ..., 100 |
| 3 | Minimum number of samples | 40 |
| 4 | Training samples | 42% of records |
| 5 | Testing samples | 58% of records |

The sampling of the collected data is shown in Table 1. Different distinct and constant principles are taken into account in sustainable environment and global citizenship education. The prediction accuracy of the proposed definition is in line with the C-ESD, AGCE, E-GC, TGCE, and AME-SD methods already outlined in the literature. The following criteria were calculated using text analysis, statistical analysis, performance analysis, predictive analysis, and discriminant analysis of the proposed system. The experimental documents collected approximately 4245 forms in six features of four grades.

### 4.1. Text Analysis

As shown in Figure 6, the analysis of a text is about scanning texts to extract facts that are readable by a computer. Text analysis is designed to generate structured data from free text. The procedure can be considered to slice and decorate unstructured, heterogeneous material into datasets that are easily managed and interpreted. Text analysis can identify a core concept of the text and analyze how it evolves while using a written strategy (a literary aspect or technique used in literature or rhetoric). To help with the analysis, using a solid and detailed text proof is recommended. Through Equation (5), the text analysis techniques have been improved.

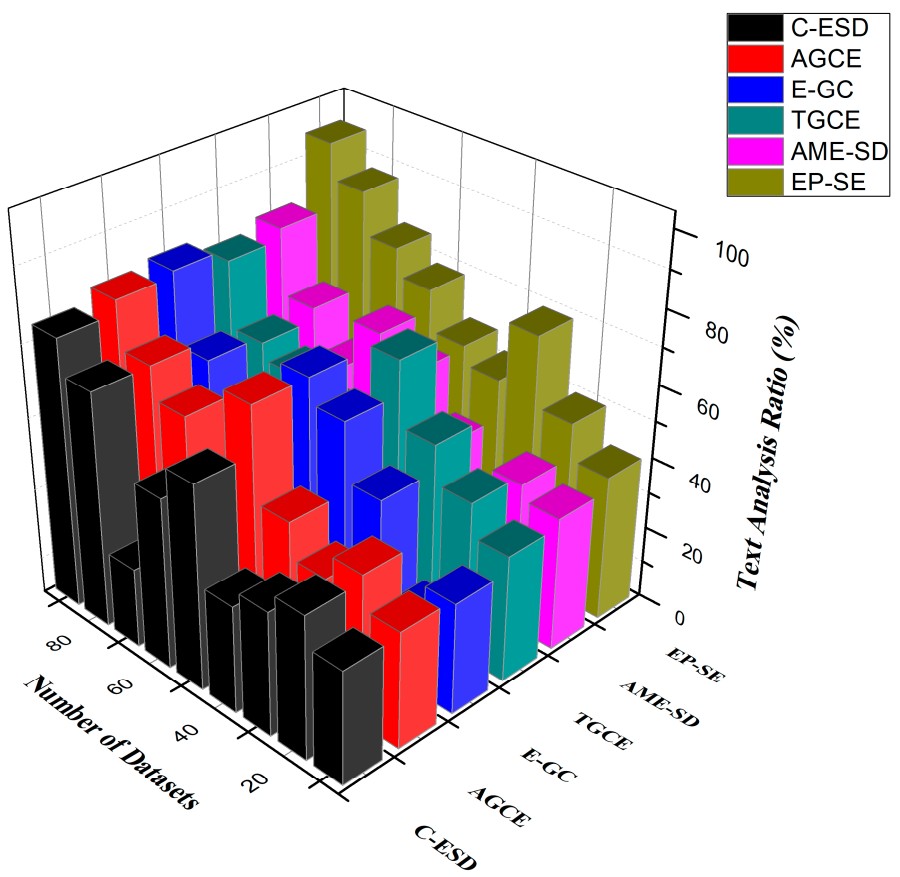

**Figure 6.** Text analysis.

*4.2. Statistical Analysis*

The science of data collection and uncovering patterns and trends in statistical research is yet another way to say "statistics". Statistics can evaluate data, after collection, as shown in Figure 7. Relevant statistical techniques for a sample of data are used for statistical analysis to evaluate the significance of the data to the entire population. It allows the generalization of data from individual markets, cohorts, and groups so that the behaviors and features of others can be theoretically predicted. Statistical analysis techniques have been improved using Equation (6).

Table 2 shows the mechanisms suggested for implementing a unique, sustainable education environment and global citizenship system. The accuracy rating results showed that the statistical ratios for C-ESD, AGCE, E-GC, TGCE, AME-SD, and EP-SE were 80.66%, 68.15%, 81.57%, 74.1%, 76.98%, and 97.87%, respectively. The EP-SE strategy increases analysis precision by 97.87% and is a better solution.

**Table 2.** Comparison of statistical analyses.

| No. of Dataset | C-ESD | AGCE | E-GC | TGCE | AME-SD | EP-SE |
|---|---|---|---|---|---|---|
| 10 | 19.78 | 21.21 | 24.54 | 18.76 | 28.76 | 32.19 |
| 20 | 21.78 | 12.65 | 45.36 | 73.56 | 32.45 | 40.88 |
| 30 | 18.65 | 26.33 | 53.66 | 48.23 | 46.76 | 49.65 |
| 40 | 32.15 | 18.98 | 35.97 | 53.26 | 30.12 | 58.89 |
| 50 | 28.56 | 36.78 | 61.78 | 31.03 | 51.34 | 53.44 |
| 60 | 45.89 | 31.87 | 49.74 | 45.69 | 67.87 | 69.45 |
| 70 | 38.27 | 56.21 | 71.23 | 77.89 | 50.65 | 79.67 |
| 80 | 68.26 | 49.21 | 31.43 | 69.31 | 70.12 | 88.69 |
| 90 | 80.66 | 68.15 | 81.74 | 74.1 | 76.98 | 97.87 |

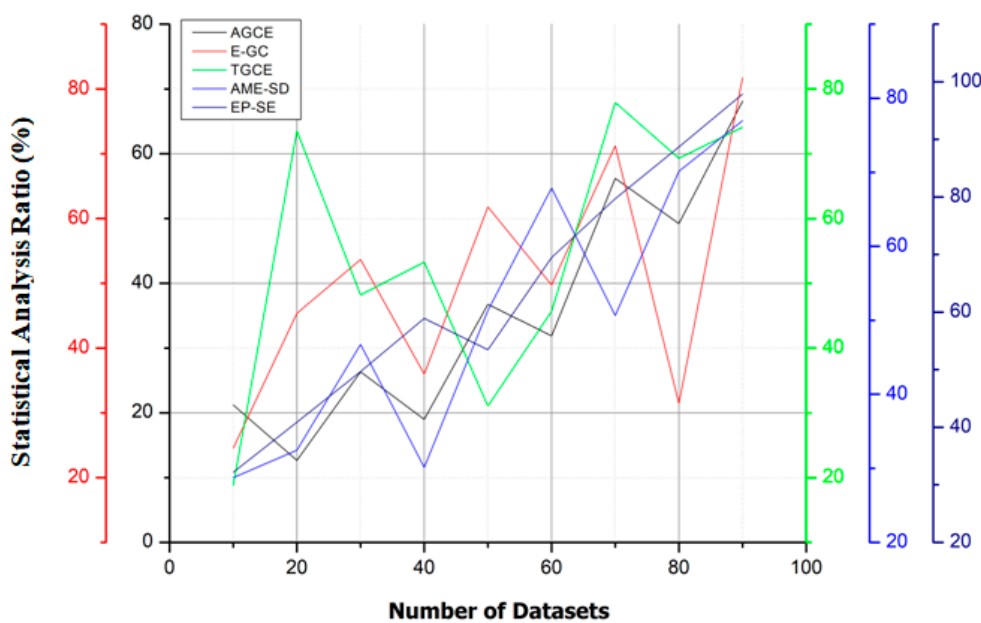

**Figure 7.** Statistical analysis.

### 4.3. Performance Analysis

Evaluation of the performance of this configuration included analytical expressions of the estimated probability distribution and the expected position value, along with the error in position prediction, which has been omitted from previous research on wired tracking devices. It can be seen in Figure 8 that this research, concentrated on a formula for coordinates based on vector data determinants, can be more conceptually traced than before because all terms are geometrically decisive. Using Equation (2), the performance analysis technique has improved.

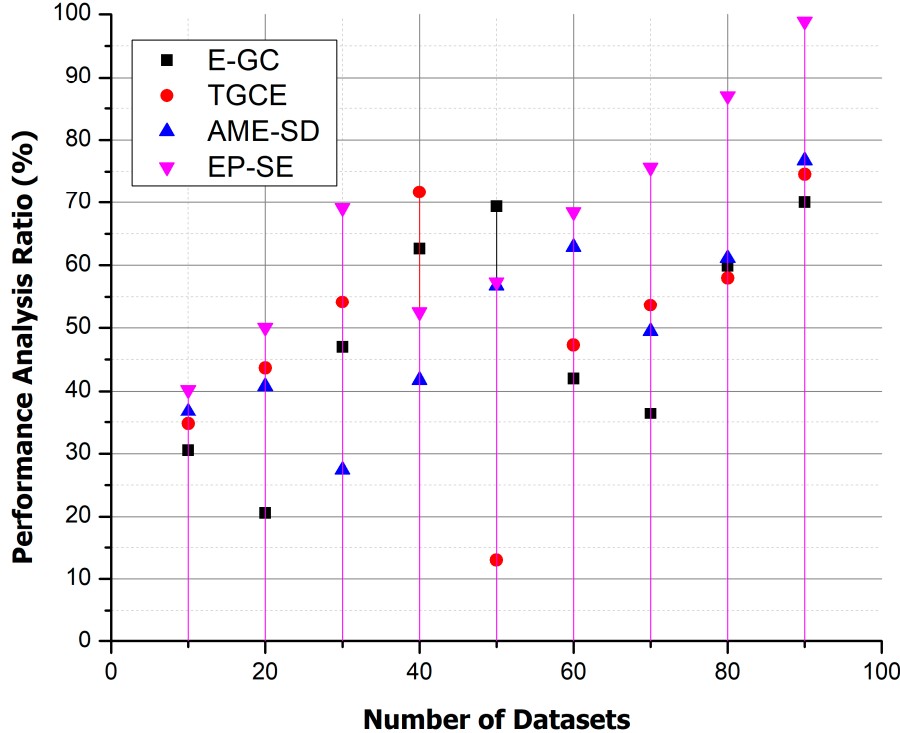

**Figure 8.** Performance analysis.

### 4.4. Predictive Analysis

Predictive analytical methods are used to identify the potential for future results based on historical information through data, statistical, and machine learning techniques. The objective is to move beyond what exists to provide a better evaluation of what is to come. Predictive analysis uses historical data to forecast events in the future. Historical data are typically used to create a statistical model that captures significant patterns. This predictive model is then used to predict the subsequent steps from current data or propose measures to achieve the best possible results, as shown in Figure 9. Using Equation (8), the predictive analysis technique has been improved.

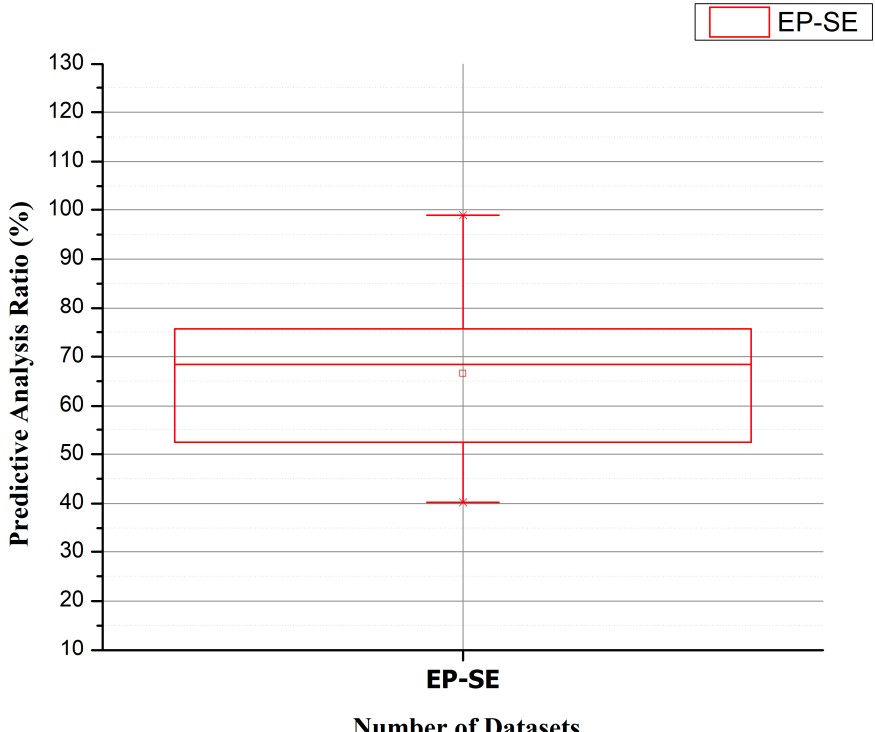

**Figure 9.** Predictive analysis.

Table 3 displays the proposed frameworks for implementing an ideal environment for sustainable education and the global citizenship scheme. The precise rating results showed that the predictive ratios were 72.66%, 69.15%, 81.57%, 74.1%, 76.98%, and 94.65% for C-ESD, AGCE, E-GC, TGCE, AME-SD, and EP-SE, respectively. The technique EP-SE improves research accuracy by 94.65% and is a better approach.

**Table 3.** Comparison of predictive analysis.

| No. of Dataset | C-ESD | AGCE | E-GC | TGCE | AME-SD | EP-SE |
|---|---|---|---|---|---|---|
| 10 | 31.33 | 32.14 | 30.56 | 34.76 | 36.76 | 40.22 |
| 20 | 39.45 | 41.24 | 20.6 | 43.65 | 40.78 | 50.01 |
| 30 | 34.15 | 34.19 | 46.99 | 54.11 | 27.45 | 69.1 |
| 40 | 29.47 | 43.76 | 62.62 | 71.65 | 41.78 | 52.43 |
| 50 | 56.33 | 69.33 | 69.34 | 12.98 | 56.78 | 57.34 |
| 60 | 47.14 | 36.54 | 41.98 | 47.26 | 62.89 | 68.44 |
| 70 | 21.89 | 56.39 | 36.41 | 53.61 | 49.45 | 75.65 |
| 80 | 65.25 | 65.25 | 59.89 | 57.98 | 61.13 | 86.98 |
| 90 | 68.78 | 70.55 | 69.99 | 74.56 | 76.77 | 98.89 |

### 4.5. Discriminant Analysis

The investigator uses discriminant analysis to evaluate the research outcomes if the criterion or variety base is definite and if the interval is between predictors or between independent variables. The analysis is sensitive to outliers and the dataset needs to be bigger than the number of predictor variables in the smaller sample. For each grouping, variable-level independent variables are natural, as shown in Figure 10. Using Equation (12a,b), the discriminant analysis technique has improved.

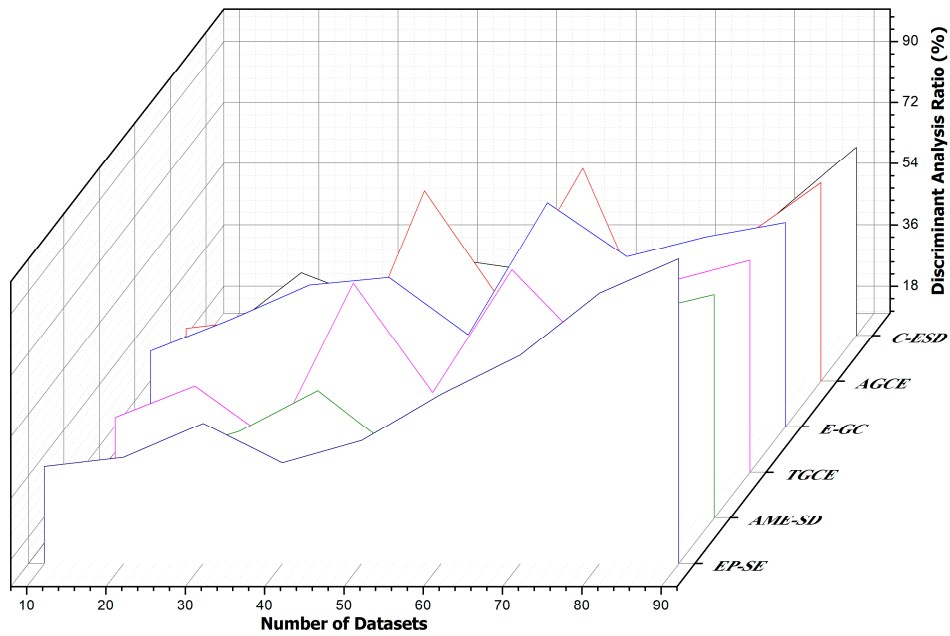

**Figure 10.** Discriminant analysis.

This research has shown the variety in education approaches and has provided non-discursive elements that focus on global citizenship activities, indicating the individual responsibilities or moral principles linked to global citizenship. Globalization based on commodity requirements can be increased through improved pedagogical sustainability.

### 5. Conclusions

The EP-SE policy and research initiative demonstrate that global civilization is not such a controversial concept. The tendency to concentrate on ordinary education for worldwide growth can be seen as an obvious answer to emergency issues in global contexts. However, general agreement is needed to prepare educated and willing people to engage in the complex world of globalization. There is not yet a global consensus. World citizenship educators must tackle and react to global disparities in internationalization and globalization, colonial legacies, and philosophies that support a structure that, although to the benefit of some, is to the detriment of many. Many educators depend on global citizenship education to open up educational facilities for a fairer, more peaceful environment.

This review of the current EP-SE policies and practices opens up claims for global citizenship. Rules that have served other educational objectives are believed to be at odds with inclusive citizenship in a world of unequal globalization. Although EP-SE systems can be said to function for justice and inclusion, these arguments conceal the basis of the program's more competitive internationalization and trade ventures. Global citizenship education must develop a broad approach, which is not always focused on shared understanding. Moreover, disciplinary, interdisciplinary, and multidisciplinary programmers can create a creative and new pedagogical place to transform the social realities of a global world and achieve their social objectives in post-secondary education. The current paper reflects the differences in education, provides non-discursive elements that represent global citizen-

ship, and suggests individual obligations or morals related to world citizenship. Increased pedagogical sustainability may lead to globalization dependent on commodities needs.

**Author Contributions:** Conceptualization, F.Z.; methodology, H.F.; software, Q.X.; validation, C.L.; formal analysis, C.L.; investigation, H.F.; resources, C.L.; data curation, Q.X.; writing—original draft preparation, H.F.; writing—review and editing, H.F.; visualization, H.F.; supervision, C.L. and F.Z.; project administration, F.Z. All authors have read and agreed to the published version of the manuscript.

**Funding:** General projects of the National Social Science Planning Fund: An Empirical Study on the Innovation of Spatial Social Governance Mechanism in Northeast China's Revitalization and Integration into the New Development Pattern [21BSH147].

**Institutional Review Board Statement:** Not applicable.

**Informed Consent Statement:** Not applicable.

**Data Availability Statement:** Not applicable.

**Conflicts of Interest:** The authors declare no conflict of interest.

**Abbreviations**

| | |
|---|---|
| EP-SE | Emerging Policies for Sustainable Environment |
| C-ESD | Context of Education Sustainable Development |
| AGCE | Advanced Global Citizenship Education |
| E-GC | Education Global Citizenship |
| TGCE | Typologies of Global Citizenship Education |
| AME-SD | Achieving and Monitoring Education Sustainable Development |
| GCD | Global Citizenship Education of Countries |
| MSE | Error Function |
| RMSE | Root Mean Square Error |

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
