# Peer review of "New Policy Research on Education Development and Global Citizenship in a Sustainable Environment"

_sustainability, doi:10.3390/su15064736_

Round 1
Reviewer 1 Report
See comments in the article

Author Response
Point 1: On page 2: why in italics? space between the words. why different fonts color and underline?
Response 1: I am very sorry that due to the author's negligence to bring you in the paper, the author carried out the italic, highlighting marks are writing problems, there is no special meaning, the author has carried out an overall check and change. The author has double-checked and revised the text for space issues.
Point 2: On page 3: see fonts. curriculum vitae??? something is wrong with the meaning…..
Response 2: The authors have revised the text in the appropriate places.
Point 3: About the picture issue. In caption do not need the point in the end of the sentence. In addition, this figure belongs to authors? If this fig. previously has published you have to identify by a reference to the original source at the end of the caption.
Response 3: In the caption, all punctuation added at the end and redundant parts were carefully checked and revised by the author. For the picture issue you mentioned. All pictures are made by the author.
Point 4: On page 3: see fonds in this paragraph and justify the text.
Response 4: The authors have made changes in the original text.
Point 5: On page 6: space.
Response 5: The authors have made changes in the original text.
Point 6: On page 7: on fonts.
Response 6: The authors have made changes in the original text.
Point 7: Numbering of the references.
Response 7: We have renumbered the references and made additions and corrections to the references.
For other parts of the original text highlighted in yellow by the reviewer, the authors have revised accordingly. The authors again apologize for any writing problems in the manuscript that caused the reviewer any trouble. In order to improve the writing level of the manuscript, the authors have applied the English editing service in MDPI to touch up and revise the paper as a whole, which hopefully will help the writing level of the manuscript.

Reviewer 2 Report
, I think it is right to make some comments on the work: the topic is certainly interesting and the introduction is well written except for the phrase The rest of the research is as follow: Section 1 Introduction of sustainable educational development and gobal citizenship
Already from section 2 the text is more difficult to understand also because of the approximation with which it was written, as evidenced by the use of different typefaces. This crudeness of the editing seems to suggest that the work was written a bit hastily.
The Figures, in particular the 2, are not understandable or well explained to a non-expert.
The transition from a discursive part to the introduction of eq.1 is quite incomprehensible to a non-expert and equally incomprehensible the following pages in which many quantities are introduced and calculated whose meaning is not clear and which are often not defined
Author Response
I think it is right to make some comments on the work: the topic is certainly interesting and the introduction is well written except for the phrase The rest of the research is as follow: Section 1 Introduction of sustainable educational development and gobal citizenship
Point 1: Already from section 2 the text is more difficult to understand also because of the approximation with which it was written, as evidenced by the use of different typefaces. This crudeness of the editing seems to suggest that the work was written a bit hastily.
Response 1: First of all, we are very sorry for the trouble caused by the non-conformity of our manuscript's writing second class. In order to remedy this problem, we have applied the English editing service in MDPI to revise the overall grammar and writing of our manuscript, which we hope will help to improve the writing level of the manuscript. Secondly, we have double-checked with the manuscript as a whole for other formatting issues, etc. We have corrected the other formatting errors, etc., in the manuscript.
Point 2: The Figures, in particular the 2, are not understandable or well explained to a non-expert.
Response 2: Thank you very much for your suggestion. We have redrawn Figure 2 for your better understanding. And we have improved the analysis of Figure 2 below.
Point 3: The transition from a discursive part to the introduction of eq.1 is quite incomprehensible to a non-expert and equally incomprehensible the following pages in which many quantities are introduced and calculated whose meaning is not clear and which are often not defined.
Response 3: Thank you very much for your suggestion. In response to your question, we have completely revised the part of the literature where we are contributing. We have also made overall changes to the writing and other issues in the article and added new references in order to better understand the content of the manuscript.
Thank you again for commenting on our manuscript. All changes in the manuscript are highlighted with a yellow marker.

Reviewer 3 Report
Manuscript ID: sustainability-2191032
Article review
Title: New Policy Research on Education Development and Global Citizenship in a Sustainable Environment
The topic of article argues an important aspect of contemporary society in which globalization processes are getting out of control. Therefore, for a sustainable future, action through the education of new generations is extremely important.
Although the paper has a good topic, the language used in the paper is too complicated and difficult to read. The paper ultimately does not bring anything particularly new that a high school student would not have written in an essay on the topic of globalization (without research).
In chaper 1. Introduction of sustainable educational development and global citizenship the authors state the contributions of the paper:
The main contribution of this paper is
(1)To reveal the educational development and provide the non-discursive components based on the operations of global citizenship.
(2)To emphasize the personal accountability and moral ideas that are identified with global citizenship.
(3)To enhance the educational sustainability that is existed to increase globalization based on the commodity condition.
The contributions elaborated in this form are too general and have no added novelty to the research field. To cover this the authors put in the paper a confused methodology with equations and factors that in the end did not produce better scientific contributions.
I think that the work is not at the level of a scientific contribution that would justify publication in the scientific Q1 journal.
I fully agree with the author's motive that it is necessary to act and moderate the globalization process and that the education of new generations is crucial in this.
I do not see in this work the quality of academic discourse or the methodology that would justify the publication. I think the topic and paper are at the level of a scientific conference and give a good basis for a panel discussion, but I do not see the relevant novelty and scientific contribution of the paper.
Therefore, I recommend that the editors reject the paper for publication in Sustainability and recommend a more suitable medium for publication and professional action to the authors.
Kind regards
Author Response
The topic of article argues an important aspect of contemporary society in which globalization processes are getting out of control. Therefore, for a sustainable future, action through the education of new generations is extremely important.
Although the paper has a good topic, the language used in the paper is too complicated and difficult to read. The paper ultimately does not bring anything particularly new that a high school student would not have written in an essay on the topic of globalization (without research).
Point 1: In chaper 1. Introduction of sustainable educational development and global citizenship the authors state the contributions of the paper:
The main contribution of this paper is
(1)To reveal the educational development and provide the non-discursive components based on the operations of global citizenship.
(2)To emphasize the personal accountability and moral ideas that are identified with global citizenship.
(3)To enhance the educational sustainability that is existed to increase globalization based on the commodity condition.
The contributions elaborated in this form are too general and have no added novelty to the research field. To cover this the authors put in the paper a confused methodology with equations and factors that in the end did not produce better scientific contributions.
Response 1: For the contribution issue you mentioned, we have completely revised the author contribution of the article.
Point 2: I fully agree with the author's motive that it is necessary to act and moderate the globalization process and that the education of new generations is crucial in this.
I do not see in this work the quality of academic discourse or the methodology that would justify the publication. I think the topic and paper are at the level of a scientific conference and give a good basis for a panel discussion, but I do not see the relevant novelty and scientific contribution of the paper.
Response 2: In order to compensate for the writing level of the article, the authors applied the English editing service in MDPI to revise the whole article for writing and grammar, which hopefully will help the writing level of the paper. At the same time, the authors made comprehensive corrections to other errors in the article such as formatting. They also added the participating literature. All changes in the article are highlighted in yellow. Thank you very much for your comments on the article, and we hope that our changes will lead to an improved article.

Round 2
Reviewer 2 Report
The authors revised the manuscript according the referee's suggestions
Reviewer 3 Report
Dear colleagues,
I still think that the paper is not at the appropriate level for a journal in Q1, but if the editor decides to publish the article, I agree with his decision.
Kind regards